# Novel Approach for the Approximation of Vitamin D_3_ Pharmacokinetics from In Vivo Absorption Studies

**DOI:** 10.3390/pharmaceutics15030783

**Published:** 2023-02-27

**Authors:** Grzegorz Żurek, Magdalena Przybyło, Wojciech Witkiewicz, Marek Langner

**Affiliations:** 1Faculty of Pure and Applied Mathematics, Wrocław University of Science and Technology, 51-270 Wrocław, Poland; 2Lipid Systems sp. z o.o., 54-613 Wrocław, Poland; 3Department of Biomedical Engineering, Faculty of Fundamental Problems of Technology, Wrocław University of Science and Technology, 51-270 Wrocław, Poland; 4Research and Development Centre, Specialized Hospital in Wrocław, 51-124 Wrocław, Poland

**Keywords:** cholecalciferol, vitamin D_3_, liposomes, pharmacokinetics, absorption rate

## Abstract

The changing environment and modified lifestyles have meant that many vitamins and minerals are deficient in a significant portion of the human population. Therefore, supplementation is a viable nutritional approach, which helps to maintain health and well-being. The supplementation efficiency of a highly hydrophobic compound such as cholecalciferol (logP > 7) depends predominantly on the formulation. To overcome difficulties associated with the evaluation of pharmacokinetics of cholecalciferol, a method based on the short time absorption data in the clinical study and physiologically based mathematical modeling is proposed. The method was used to compare pharmacokinetics of liposomal and oily formulations of vitamin D_3_. The liposomal formulation was more effective in elevating calcidiol concentration in serum. The determined AUC value for liposomal vitamin D_3_ formulation was four times bigger than that for the oily formulation.

## 1. Introduction

Vitamins are important for maintaining human health and wellbeing. Some of them are available only from exogenous sources as exemplified by vitamin C or vitamin B_12_. Since they are not produced endogenously, the determination of their pharmacokinetics is a straightforward task, at least from the methodological point of view [1,2]. The pharmacokinetic curve of vitamin C can be derived from a short time clinical study. [3,4]. Vitamin D_3_ is different. It originates from two sources: skin and foodstuff. This and the long half-time (about 20 days) make the determination of its pharmacokinetics challenging [5]. The most direct approach is to use a radiolabeled compound [6]. Regardless of the method used, there are numerous logistic and methodological difficulties making large-scale clinical studies demanding. In the paper, we propose a new method for vitamin D_3_ pharmacokinetic curve determination. It combines semi-physiological modeling and short time absorption data from clinical studies.

Vitamin D_3_ is a compound, of which a deficiency can lead to numerous health risks [7]. Its importance for human organism functioning is demonstrated by the fact that the vitamin is required for the expression of over 700 genes [8,9]. Vitamin D_3_ is also critical element of calcium homeostasis [10]. The altered lifestyle of modern man (little exposure to natural sunlight) and changes in nutritional habits has resulted in wide-spread vitamin D_3_ deficiency [7]. Therefore, effective modes of supplementation are an important element of preventive medicine [11].

Vitamin D_3_ (cholecalciferol) is a precursor for its physiologically active metabolites calcidiol and calcitriol. The hydrophobic character of vitamin D_3_ and its metabolites requires complex absorption and redistribution mechanisms [2,12]. This is because hydrophobic compounds can be transferred across aqueous spaces only when associated with other molecule(s), forming water-soluble aggregates. This is critical since the cholecalciferol absorption (intestine) or synthesis (skin) are located in distal places with respect to the location of the first and the second steps of its metabolic transformation into the active form (liver and kidney). The transfer of hydrophobic compounds between the two locations requires mechanisms relying on dedicated and/or nonspecific carriers in the form of proteins and/or lipoproteins. Given orally, the cholecalciferol absorption consists of the sequence of events facilitated by ill-defined aggregates dispersed in the aqueous phase of the gastrointestinal tract. Vitamin D_3_ can reach the intestine wall only when it is in the aggregate capable to cross the mucus layer [12]. When vitamin D_3_ reaches enterocytes, it is internalized and released into serum while associated with specific carrier proteins (VDBP) [13,14] or with lipoproteins [15,16]. Its subsequent enzymatic transformation in the liver and kidneys is possible only because of this distribution system. Transport of vitamin D_3_ and its metabolites is controlled by proteins and/or lipoprotein mediators.

It can be assumed that vitamin D_3_ internalization, redistribution, and subsequent transformation to calcidiol (25(OH)D_3_), which is routinely used for diagnostic purposes [17], can be qualitatively divided into two phases: before and after the internalization by competent cells in the intestine. Events prior to internalization by enterocytes are characterized by low specificity, whereas subsequent biodistribution and metabolic transformations are facilitated by very specific intracellular and extracellular events [9]. Absorption in the intestine can be greatly modified by changing the vitamin D_3_ formulation. [18,19]. This is because hydrophobic vitamin D_3_ is not capable of reaching the surface of the enterocyte alone. Instead, the absorption of vitamin D_3_ consists of a sequence of physicochemical and enzyme-assisted transformations taking place in the gastrointestinal tract [20]. Because water insoluble crystals of vitamin D_3_ cannot be effectively absorbed, it is commonly given as an oil formulation [15,21,22]. This by itself does not ensure efficient absorption since dispersion of oil in the aqueous phase will form an unstable and heterogenous emulsion [23,24]. For example, the coalescence of oily suspension will affect the digestion processes [12,25]. To be internalized, droplets of oil with vitamin D_3_ should reach the surface of the enterocytes [26,27]. This happens only when oily droplets can cross the mucus layer containing polymeric mesh formed by mucins. The mucus serves as a filter, preventing large particulates (>400 nm) from reaching enterocytes [28]. Vitamin D_3_ can be absorbed only when dissolved in particulates which are able to pass the mucus layer. This is difficult to achieve using the oily suspension alone [28,29]. All that will result is the limited capacity of an oily formulation to facilitate the efficient absorption of vitamin D_3_ [22,30]. Liposomes are convenient and biocompatible oral delivery vehicles [16,22]. They easily accommodate the hydrophobic vitamin D_3_ within their lipid bilayer and their size can be tailor-made. In addition, the mechanical stability of liposome lipid bilayer ensures that the cargo is delivered to the intestine wall ready for absorption [31,32]. In addition, when exposed to conditions simulating stomach and intestine environments (pH, temperature, bile salts and enzymes), their topology remains unchanged for a sufficiently long time to ensure the effective absorption. The main obstacle for the application of liposomes in everyday products has been the limited scale of available production processes and the belief that they are inherently unstable [33]. Those difficulties have been resolved by the introduction of Liposhell^®^ technology, which is capable of delivering large quantities (tones) of a homogenous suspension of stable liposomes encapsulating a wide variety of active ingredients, including proteins and nucleic acids. The absorption of vitamin D_3_ in liposomes relies on their endocytosis and trans-endocytosis by enterocytes and possible M cells [6,34]. In addition, the stability of liposomes in the suspension enhances the internalization, regardless of the accompanying foodstuff. Table 1 shows characteristic properties of the two formulations of vitamin D_3_.

The liposome suspension produced using Liposhell^®^ technology is characterized by well-defined population of liposomes, ensuring optimal conditions for their internalization. The Liposhell^®^ technology is based on the formation of tightly packed, uniform with respect to size, lipid vesicles (liposomal gel). The design of the liposome-based delivery strategy assumes that vitamin D_3_ is located in the hydrophobic core of the lipid bilayer. The stability of the liposome-vitamin D_3_ system with the presence of lipid vesicles and albumins was evaluated previously using isothermal titration calorimetry [38]. Specifically, no significant thermal signal was observed when liposomes with vitamin D_3_ were added to vitamin D_3_-free liposomes or albumins, showing that vitamin D_3_ does not equilibrate between liposomes and albumin, nor between liposomes. From the experiment, it has been concluded that, during digestion, vitamin D_3_ remains in liposomes regardless of their fate in the gastrointestinal tract.

## 2. Materials and Methods

### 2.1. Materials

Soybean phosphatidylcholine (Phospholipon 90G) and vitamin D_3_ (cholecalciferol) were purchased from Lipoid GmbH (Ludwigshafen, Germany) and DSM Nutritional Product Ltd. (Village Neuf, France), respectively. Chloroform and methanol were obtained from VWR (Radnor, PA, USA), whereas ethanol and NaOH were from Stanlab (Lublin, Poland). All solutions were prepared with commercially purified water (AquaEngineering, Warsaw, Poland).

### 2.2. Preparation of Liposomal Vitamin D_3_ Formulation

Liposomal vitamin D_3_ formulation was prepared as described in details elsewhere [36]. In short, liposomes were formed by mixing the organic phase with the aqueous phase (1:1 *w*/*w*). The organic phase contains propylene glycol, phospholipids (20% *w*/*w* in the final preparation), and vitamin D_3_. Next, the viscous gel was extruded through the 100 nm polycarbonate filter (Nucleopore Corp., Pleasanton, CA, USA). The content of phosphatidylcholine was determined by the HPLC-ELSD method [3]. Finally, the lipid gel was diluted 200 times with the glycerol/water (1:1 *w*/*w*) mixture and supplemented with natural flavor (0.3 *w*/*w*) and pectin (2.45 *w*/*w*). Liposomal vitamin D_3_ was prepared by Lipid Systems Sp. z o.o. (Wrocław, Poland) under conditions satisfying HACCP and GMP requirements.

### 2.3. Characterization of Liposomal Vitamin D Formulation

The size distribution of liposomes in the liposomal vitamin D_3_ formulation was determined by the dynamic light scattering method with some modifications in the preparation of the measured samples due to the presence of pectin (Zetasizer Nano ZS, Malvern, UK). The quantity of vitamin D_3_ was determined with RP-HPLC (Reversed-Phase High Liquid Chromatography) according to the method developed by Sazali et al. [39], with some modifications. The modular HPLC set composed of a pump (Azura P4.1S, KNAUER, Berlin, Germany), autosampler (Marathon Basic, Spark Holland, Emmen, The Netherlands), peltier column thermostat (Jetstram II Plus, Knaure, Berlin, Germany), and a UV-VIS detector (Azura UVD 2.1L, KNAUER, Berlin, Germany) was used. The separation was achieved using: 4.6 × 250 mm; 5 μm particles, 100 Å pore sizes, column (Eurospher 100-5 C18, KNAUER, Berlin, Germany). The freshly prepared mixture of methanol and water (98:2 *v*/*v*) as the isocratic mobile phase was pumped at a flow rate of 1 mL/min at 40 °C. The injection volume was 20 μL. Samples for the calibration curve were prepared at the concentration range of 0.6–9 μg/g of vitamin D_3_ in ethanol. Samples of liposomal formulations were dissolved in ethanol at the 4/10 (*w*/*w*) ratio, mixed, centrifuged (2800 rpm for 10 min.), and filtered through 0.2 μm cellulose membrane before analysis [40].

### 2.4. Cryogenic Transmission Electron Microscopy (TEM) Imaging

Cryogenic Transmission Electron Microscopy (cryo-TEM) images were collected with a Tecnai F20 X TWIN microscope (FEI Company, Hillsboro, OR, USA) equipped with a field emission gun operating at an acceleration voltage of 200 kV. Images were recorded on the Gatan Rio 16 CMOS 4 k camera and processed with Gatan Microscopy Suite (GMS) software (Gatan Inc., Pleasanton, CA, USA). Specimen preparation was done by the vitrification of the aqueous solutions on grids with holey carbon film (Quantifoil R 2/2; Quantifoil Micro Tools GmbH, Großlöbichau, Germany). Prior to use, the grids were treated for 15 s in oxygen plasma using a Femto plasma cleaner (Diener Electronic, Ebhausen, Germany). Cryo-samples were prepared by applying a droplet (3 μL) of the suspension to the grid, blotting with filter paper, and immediate freezing in liquid ethane using a fully automated blotting device Vitrobot Mark IV (Thermo Fisher Scientific, Waltham, MA, USA). The vitrified specimens were kept under liquid nitrogen prior the insertion into a cryo-TEMholder Gatan 626 (Gatan Inc., Pleasanton, CA, USA) and analyzed at −178 °C.

### 2.5. Clinical Studies and Quantification of Calcidiol in Serum

The clinical experiment has been thoroughly described elsewhere [36]. In summary, the study was performed on 18 healthy volunteers (age 24–65) according to “the cross-over design”. Following a 12 h fasting, each volunteer was given 10,000 IU of vitamin D_3_, either in the liposomal or oily formulation. After 3 weeks, the experiment was repeated, but volunteers consumed the other vitamin D_3_ formulation. The marketed product was used as the oily formulation. The quantity of vitamin D_3_ in oily formulation was used as specified by the producer. Less than 50 μL of blood was drawn from a finger of each volunteer at eight-time points; the reference sample shortly before and 0.5, 1, 1.5, 2, 3, 4, and 5 h following the intake of vitamin D_3_. Blood samples of the volunteer were collected by qualified personnel, and the concentration of 25(OH)D_3_ measured by Cambridge Diagnostics Sp. z o.o. (Poland). All procedures involving humans were approved by the Bioethical Commission at the Research and Development Centre at the Specialized Hospital in Wrocław, number: KB/07/2020. The vitamin D_3_ absorption was evaluated based on the concentration of calcidiol in serum. Specifically, the initial quantity of 25(OH)D_3_ (shortly before supplementation, A_0_) was subtracted from its values at later time points, A(t), and the obtained difference normalized to the initial value [(A(t) − A_0_)/A_0_]. Next, the tendency of 25(OH)D_3_ change was approximated with the linear function using the last square fitting. A straight line was then used to estimate the calcidiol concentration in serum for the pharmacokinetic curve reconstruction.

### 2.6. The Reconstruction of Pharmacokinetic Curve for Calcidiol

The absorption of vitamin D_3_ is followed by its transfer to the liver, where it is metabolically transformed. The literature data show that the pharmacokinetic curve for calcidiol reaches a maximum at three days after supplementation with cholecalciferol. However, the steep rise of calcidiol concentration in serum shortly after supplementation indicates rapid absorption and enzymatic transformation of cholecalciferol to calcidiol in the liver. The shape of the pharmacokinetic curve during the first day after supplementation justifies the assumption that calcidiol can be evaluated even during the first few hours following supplementation. The advantage of such approach is the elimination of possible interferences resulting from unpredictable and difficult to control volunteer behavior, when outside the medical facility (exposure to the sun and/or variation in diets). Consequently, during the reduced time of the experiment, the observed rise of calcidiol is affected exclusively by the absorption efficiency. This is because, after fasting, the supplement will pass the absorption zone in the intestine during the few hours following the supplementation [41].

When a single dose of vitamin D_3_ is administered, its quantity in serum can be described by the following formula:(1)$$\left[D_{3}\left(t\right)\right]=-k\int_{0}^{t}\left[D_{3}\left(t\right)\right]dt$$
where [*D*_3_(*t*)] represents the concentration of vitamin D_3_ in serum. The amount of vitamin D_3_ absorbed in the intestine [*D*_3_(0)] and can be expressed by the empirical formula [29]:(2)$$\left[D_{3}\left(0\right)\right]=\int_{0}^{t_{transit}}J_{muc}\left(t\right)dt$$
where $t_{transit}$ stands for the time, when vitamin D_3_ resides in the region of intestine where the absorption is taking place, and $J_{muc}\left(t\right)$ represents the flux of vitamin D_3_ across mucus lining the intestine wall. Parameters which define the absorbed quantity—$t_{transit}$ and $J_{muc}\left(t\right)$—depend on vitamin D_3_ formulation. When formulation disperses uniformly in the stomach content, the transit time will increase. No such effect will be observed when the coalescence occurs [16]. The flux of vitamin D_3_ across the mucus will depend predominantly on properties of particulates such as surface charge, propensity for enzymatic activity, ability to accommodate bile acids, and most importantly their size. The size of particulates is an important parameter since mucus can be penetrated only when it does not exceed 400–500 nm [26,28].

The quantity of 25(OH)D_3_ in serum can be approximated by the following equation:(3)$$\left[25\left(OH\right)D_{3}\left(t\right)\right]=\int_{0}^{t}\left\{k\left[D_{3}\left(t\right)\right]-m\left[25\left(OH\right)D_{3}\left(t\right)\right]\right\}dt$$

$\left[D_{3}\left(t\right)\right]$ is a cholecalciferol concentration, $\left[25\left(OH\right)D_{3}\left(t\right)\right]$ stands for the concentration of calcidiol, and *k* and *m* are constants describing the synthesis (liver) and degradation (kidneys) of calcidiol. It has been assumed that the quantity of vitamin D_3_ at t = 0 equals to the amount of vitamin D_3_(0) absorbed. The dependence of a calcidiol concentrations on time follows the formula:(4)$$\left[25\left(OH\right)D_{3}\left(t\right)\right]=\frac{ck}{m}+\frac{ke^{-kt}}{m-k}+\alpha e^{-mt}$$

Except the [*D*_3_(0)] value, all other parameters are dependent on metabolic processes. To calculate those parameters, the experimental data from the literature [42] were normalized and the function (4) was optimized with respect to k and m, assuming that the initial calcidiol concentration C(0) = 0. In addition, we assumed that, for a single application, c = 0 since $\log_{t\to \infty }C\left(t\right)=0$. The optimalization of the dissipation function equals the average distance between experimental and theoretical points. Consequently, the pharmacokinetic curve of calcidiol can be reconstructed from the combination of time-limited clinical studies and mathematical equations derived from experimental data [42].

## 3. Results and Discussion

The absorption of vitamin D_3_ in liposomes is most efficient when their size does not exceed 250–300 nm [43]. Figure 1 shows the size distribution of liposomes along with the respective correlation function determined using dynamic light scattering. The average size of liposomes and polydispersity index (PDI) were equal to 117 nm. and 0.23, respectively. Visualization of liposomes with cryoTEM demonstrated that they are spherical, unilamellar, and confirmed the homogeneity of their size distribution. When an oily formulation of vitamin D_3_ is dispersed in water, it forms a heterogenous emulsion. The two formulations are expected to behave differently in the gastrointestinal tract, affecting the absorption efficiency of vitamin D_3_. The short clinical experiment showed that the measured absorption rate, evaluated with slopes of linear functions, were statistically different and equaled to 7.03 × 10^−5^ 1/min ± 3.02 × 10^−4^ 1/min and 4.01 × 10^−4^ 1/min ± 7.12 × 10^−4^ 1/min for oily and liposomal formulations, respectively [35]. Those values were used to calculate the calcidiol concentration in serum at $t_{transit}$. The developed method was next used in studies regarding characterization of two vitamin D_3_ formulations. In the analysis it had been assumed that the quantity of cholecalciferol and calcidiol in serum were at a concentration range where all proteins involved in cholecalciferol absorption, distribution, and metabolic transformation were not saturated. This means that the quantities of the absorbed cholecalciferol and the concentration of calcidiol in serum can be quantitatively correlated. Examples of pharmacokinetic curves derived for persons with vitamin D_3_ deficiency for two different formulations (oily and liposomal) are presented on Figure 2.

The reconstructed pharmacokinetic profiles of the oily and liposomal formulations were assessed using the standard single-compartmental pharmacokinetic analysis. Using the methodology, the following pharmacokinetic parameters have been derived: the maximum concentration of calcidiol (C_max_), the area under curve for the first day following the dose (AUC_1day_), the area under the curve calculated from the 2nd to the 30th day following the dose (AUC_2day-30days_), and the AUC_1day-30day_—the area under the plasma concentration–time curve integrated from zero to 30 days following the supplementation (AUC_1day-30day_), calculated as a sum of AUC_1day_ and AUC_2day–30 day_. The time at which the maximum concentration is reached (T_max_) and the elimination half-life of calcidiol (t_1/2_) depend exclusively on metabolic processes, meaning that they do not depend on the type of formulation [42] (Table 2).

The pharmacokinetics of calcidiol following the supplementation with cholecalciferol depends on the vitamin D_3_ formulation. Clinical experiment has shown that for persons with a significant deficiency of vitamin D_3_, the application of liposomal formulation causes a significant and rapid increase of calcidiol concentration in serum. The effect of the oily formulation of cholecalciferol is much smaller. Specifically, AUC_1day_ determined for calcidiol, reflecting the efficiency of cholecalciferol absorption, increased from 0.037 [day·ng/L] to 0.135 [day·ng/L], almost 3.6 times, which makes the increase of the AUC_1day-30day_ from 1.69 [day·ng/L] to 8.5 [day·ng/L], i.e., 5 times. The AUC_1day-30day_ relates to pharmacokinetics of calcidiol, which depends on the quantity of absorbed cholecalciferol and physiological processes affecting its serum concentration. The increase in the maximum concentration of calcidiol is also substantially elevated from 0.067 [day·ng/L] to 0.335 [day·ng/L]. Derived numbers confirmed the prediction that liposomes with well-defined sizes are a far more effective formulation in delivering hydrophobic vitamin D_3_ then their oily equivalent.

## 4. Discussion

The vitamin D_3_ digestion and absorption processes depend on physiological factors such as low pH and mechanical agitation in the stomach, followed by enzymatic degradation and bile acid emulsification in the intestine. In addition, the digestion will be affected by the presence of foodstuffs and the type of vitamin D_3_ formulation [12]. The two formulations of cholecalciferol (oil and liposomes) selected for comparison are qualitatively different. Since the formation of an emulsion from the oily formulation is taking place in the gastrointestinal tract, it is impossible to control, and the resulting dispersion of oil will be heterogenous with propensity to coalesce [26,29]. The suspension of liposomes is different; it is a two-phase system, where the hydrophobic region of the spherical lipid bilayer separates two aqueous phases. Liposomes can be produced in such a way that they will form a suspension, which is uniform with respect to the liposome size. In such a system, cholecalciferol is in the hydrophobic region of a liposome-forming lipid bilayer [36]. Liposomes are very stable structures, which cannot be easily destabilized unless their chemical composition is altered. This can be achieved by the action of digestive enzymes or by an association with natural detergents (bile acids) [31,32]. Importantly, dilution or mixing of liposome suspension with foodstuff will not affect their topology, nor their size. The uniform distribution of liposomes within foodstuff results in the extended exposure time. In addition, with appropriate sizes (less than 400 nm), liposomes can enter body compartments by two independent pathways: endocytotic and transendocytotic. Whereas the pathway from endosome to chylomicrons is complicated and time-consuming [44], transcytosis will result in intact liposomes crossing the intestine wall and rapidly entering the immunological system [27]. The latter will accelerate the appearance of cholecalciferol in serum and the subsequent transformation to calcidiol.

The oily formulation behaves differently; it may coalesce in the stomach, shortening the exposure time to enterocytes. The additional factor affecting the absorption of vitamin D_3_ from the oily formulation is its high heterogeneity and low stability upon digestion. In summary, whereas liposomes will remain unchanged or decrease in size, the sizes of oily droplets, characterized by their propensity to coalescence, will increase [35]. With the much shorter residence time in the intestine and a non-optimal size distribution, only aa fraction of oil (vitamin D_3_) will have access to the surface of enterocytes. The vitamin D_3_ redistribution mechanisms and its transformation to calcidiol in the liver make it difficult to alter metabolic processes. Consequently, it can be assumed that the quantity of calcidiol will scale with the amount of absorbed vitamin D_3_. The quantity of absorbed vitamin D_3_ from a dose will depend on the exposure time and formulation-dependent accessibility for competent cells. The exposure time is a function of peristaltic activity, stomach content, and the distribution of vitamin D_3_. Clinical data presented by Armas et al. [42] shows the pharmacokinetics of radiolabeled vitamin D_2_ and vitamin D_3_ following oral administration. The dependence of concentration profiles in the absorption phase of the two formulations are very similar. Dependence of the concentration of calcidiol as a function of time is different. The maximum is reached only on the 5th day for vitamin D_2_ and on about the 15th day for vitamin D_3_. These data show that the absorption and biodistribution processes can be analyzed separately and that the character of calcidiol pharmacokinetic curves will be affected by the quantity of absorbed vitamins and the efficiency of the subsequent metabolic processes. Rapid absorption of vitamin D_3_ justifies the approach, in which the initial rise of calcidiol serum concentration is driven predominantly by the quantity of absorbed vitamin D_3_. Later, when the absorption phase is completed, the pharmacokinetic profile of calciferol depends exclusively on the metabolic processes [8]. Therefore, the shape of the pharmacokinetic curve is preserved, but the value of AUC will depend exclusively on the quantity of the absorbed vitamin [12,20,45,46,47]). Consequently, the pharmacokinetic curve for a formulation can be reconstructed from the quantity of absorbed vitamin D_3_ and the predetermined shape of the pharmacokinetic curve [1,48,49].

## 5. Conclusions

The effective supplementation of hydrophobic vitamin D_3_ requires the application of excipients, which will facilitate its dissolution. However, this by itself will not ensure its effective absorption. The efficient vitamin D_3_ absorption will take place from formulations in the form of droplets characterized by sizes smaller than 250–300 nm, resulting from the size of the intestine mucus pores [43]. Dispersion of liposomes with vitamin D_3_ is an example of such a formulation. The absorption of vitamin D_3_ in liposomal and oily formulations was evaluated using short-term absorption data supplemented with physiology-based mathematical modeling. The quantity of absorbed vitamin D_3_ in vivo was estimated with the increased concentration of calcidiol in serum, whereas the shape of the pharmacokinetic curve was reconstructed with a physiologically based mathematical model. The analysis shows that the liposomal vitamin D_3_ formulation is more effective than its oily solution.

## Figures and Tables

**Figure 1 pharmaceutics-15-00783-f001:**
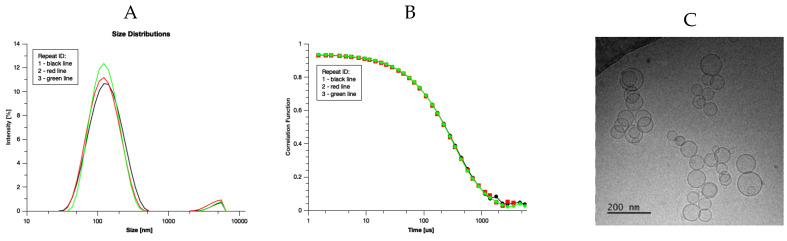
Size distribution of liposomes with vitamin D_3_. Panel (**A**) shows the liposome size distributions. Panel (**B**) presents correlation functions. The average diameter of liposomes and polydispersity index were equal to 117 nm and 0.23, respectively. Panel (**C**) presents the cryoTEM image of liposomes with vitamin D_3_.

**Figure 2 pharmaceutics-15-00783-f002:**
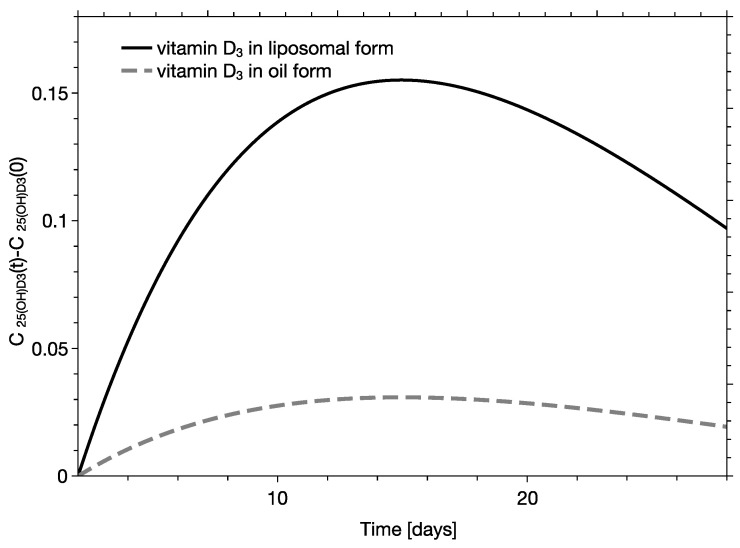
Example of the dependence of the increase of calcidiol concentration in serum ($C_{25\left(OH\right)D3}\left(t\right)-C_{25\left(OH\right)D3}\left(0\right))$ on time, as determined for a randomly selected person characterized by a low initial calcidiol concentration. The continuous and dotted curves represent the liposomal and oily formulation, respectively. The concentration of calcidiol in serum is in ng/L.

**Table 1 pharmaceutics-15-00783-t001:** Properties of liposomal and oily vitamin D_3_ formulations relevant for the absorption in the intestine [15,18,19,27,35,36,37].

	Liposome	Oil
The outcome of the homogenization in stomach	predictable	unpredictable
Stability of the dispersion with respect to droplet size	high	low
Uniformity of droplet population	high	low
Stable droplet size distribution during digestion	medium	low
Ability to cross the mucus barrier	high	limited

**Table 2 pharmaceutics-15-00783-t002:** Examples of quantitative values of pharmacokinetic parameters determined for a person with significant vitamin D_3_ deficiency.

Parameter	Liposome Formulation	Oil Formulation
C_max_	0.335 [ng/L]	0.067 [ng/L]
T_max_	14 days	14 days
t_1/2_	43 days	43 days
AUC_1day_	0.135 [day·ng/L]	0.027 [day·ng/L]
AUC_2days-30day_	8.646 [day·ng/L]	1.718 [day·ng/L]
AUC_1day-30day_	8.511 [day·ng/L]	1.692 [day·ng/L]

## Data Availability

Not applicable.

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
