# Peer review of "Novel Approach for the Approximation of Vitamin D3 Pharmacokinetics from In Vivo Absorption Studies"

_pharmaceutics, 2023, doi:10.3390/pharmaceutics15030783_

Round 1

Reviewer 1 Report

Authors describes “Novel approach for approximation of the vitamin D3 pharmacokinetics from the in vivo absorption studies”. The study is very short and various parameters did not performed. So, I will recommend article after resolving following points.

1.       Authors are advised to improve abstract in brief. Add some data in abstract too.

2.       In Materials and Methods, Material/chemicals section missing. Add this section.

3.       Line 141- Why the content of phosphatidylcholine was determined by HPLC-ELSD method?

4.       Line 179- mention the volume of Blood withdrawn.

5.       Author should check the stability of liposome at  4°C, room temperature (25 °C), and physiologic temperature (37°C).

6.       Tmax and t1/2 are same for both liposome and lipid formulation. Need to discuss.

7.       Authors are advised to add asterics (*) of statistics in Table 1.

8.     Please carefully check the text for writing and grammar.

Author Response

Reviewer 1

Authors describes “Novel approach for approximation of the vitamin D3 pharmacokinetics from the in vivo absorption studies”. The study is very short and various parameters did not performed. So, I will recommend article after resolving following points.

  1. Authors are advised to improve abstract in brief. Add some data in abstract too.

Our response: The abstract has been reformulated.

  1. In Materials and Methods, Material/chemicals section missing. Add this section.

Our Response: The data regarding materials used has been added.

  1. Line 141- Why the content of phosphatidylcholine was determined by HPLC-ELSD method?

Our Response: The relevant comment has been added to the text. HPLC-ELSD is a method capable to measure the presence of any impurities. This is important when formulation chemical stability is evaluated.

  1. Line 179- mention the volume of Blood withdrawn.

Our Response: The relevant information has been added.

  1. Author should check the stability of liposome at 4°C, room temperature (25 °C), and physiologic temperature (37°C).

Our Response: The relevant stability studies have been performed as required by regulating authorities. The regarding stability of liposomes has been added in the text.

  1. Tmax and t1/2 are same for both liposome and lipid formulation. Need to discuss.

Our Response: The dependence of time profile of the calcidiol concentration in serum is regulated exclusively by physiological processes, therefor the dependence of its concentration in serum on time should not change when the molecule is no longer associated with delivery vehicle. The relevant notation has been added in the text.

  1. Authors are advised to add asterics (*) of statistics in Table 1.

Our Response: Plots presented in Figure 2 and data given in Table 1 have been calculated for a single patient selected from volunteers participating in clinical studies Selected patient has significant vitamin D3 deficiency. For that reason, no statistical analysis has been performed.

  1. Please carefully check the text for writing and grammar.

Our Response:Text has been checked and detected mistakes have been corrected.

Reviewer 2 Report

1. Abstract is poorly written. Authors are advised to go through a structured abstract that contains background, methods, results and conclusion without mentioning these heading.

2. Authors are advised to remove all the expressions mentioned between practice (kindly refers to line 44, 68, 99, .....etc) and to re-rewrite these sentences in a scientific way.

3. The paper is generally suffering from many typo and grammar errors.

4. In the introduction, authors should account on the stability of the liposomes in the GIT. Also, authors must give a short brief about Liposhell technology (line 125). 

5. No materials section has been mentioned. What was the source of the materials used (city and country)?

6. The preparation method for the liposomal vitamin D3 formulation is not clear. It should be mentioned in a simple way to be easily reproducible. 

7. What was the apparatus used in the characterization of the formulation by the dynamic light scattering.

8. Authors should mention a brief about the HPLC method.

9. How many volunteers were participated in the study?

10. Line 180; What is meant by Sp. z.o.o.?

11. Line 205; authors mentioned a differential equation for the quantity of vitamin D in serum, this equation must be integrated and write in the linear form. The same for line 220.

12. What is meant by the formulation will phase separate (line 215)?

13. Figure 1 is not clear and the title for figure 1B and figure 2 must be mentioned in a more specific and clear way.

14. Units for all AUC parameters are missing in table 1.

15. Lines 290-292, Is this a general concept or something specific related to vitamin D? Please clarify and support with documents.

16. Liposomes oral administration faces great challenges, as their components are susceptible to GIT conditions, lipid digestion sets liposomes at risk of degradation before absorption through the enterocytes [We at al. Oral delivery of liposomes. Therapeutic Delivery. 6 (2015) 1239-1241]. Again, authors must account of the stability pf the prepared liposomal formulation.

17. The conclusion is very poor and must be improved. 

Author Response

  1. Abstract is poorly written. Authors are advised to go through a structured abstract that contains background, methods, results and conclusion without mentioning these heading.

Our response: The abstract has been reformulated.

  1. Authors are advised to remove all the expressions mentioned between practice (kindly refers to line 44, 68, 99, .....etc) and to re-rewrite these sentences in a scientific way.

Our response: The unnecessary expressions have been removed from the text.

  1. The paper is generally suffering from many typo and grammar errors.

Our response: The manuscript has been thoroughly checked for typos and mistakes.

  1. In the introduction, authors should account on the stability of the liposomes in the GIT. Also, authors must give a short brief about Liposhell technology (line 125).

Our response: The relevant information regarding the liposomes production process and stability have been added to the Material and Method section.

  1. No materials section has been mentioned. What was the source of the materials used (city and country)?

Our response: The materials section has been added.

  1. The preparation method for the liposomal vitamin D3 formulation is not clear. It should be mentioned in a simple way to be easily reproducible.

Our response: The description of the liposomal vitamin D3 formulation has been extended. In addition, the reference to the patent has been added.

  1. What was the apparatus used in the characterization of the formulation by the dynamic light scattering.

Our response: The name of the instrument has been added in the method section.

  1. Authors should mention a brief about the HPLC method.

Our response: The description of HPLC method has been added to the Materials and Method section.

  1. How many volunteers were participated in the study?

Our response: The number of volunteers has been added in the description of clinical studies.

  1. Line 180; What is meant by Sp. z.o.o.?

Our response: sp.z o.o. is the term describing the legal status of the company and is equivalent to the English term LLC (Limited Liability Company). The abbreviation sp. z o.o. is a part of the formal company name.

  1. Line 205; authors mentioned a differential equation for the quantity of vitamin D in serum, this equation must be integrated and write in the linear form. The same for line 220.

Our response: Equations 1 and 3 have been reformulated as suggested by the reviewer.

  1. What is meant by the formulation will phase separate (line 215)?

Our response: We agree with the reviewer that term “phase separation” is not fortunate in this case. We describe the effect by better defined term coalescence. The text was change accordingly.

  1. Figure 1 is not clear and the title for figure 1B and figure 2 must be mentioned in a more specific and clear way.

Our response: The appropriate changes have been made in the text.

  1. Units for all AUC parameters are missing in table 1.

Our response: Units have been added in the Table 1.

  1. Lines 290-292, Is this a general concept or something specific related to vitamin D? Please clarify and support with documents.

Our response: The text was changes as suggested by reviewer and suitable citation was added.

  1. Liposomes oral administration faces great challenges, as their components are susceptible to GIT conditions, lipid digestion sets liposomes at risk of degradation before absorption through the enterocytes [We at al. Oral delivery of liposomes. Therapeutic Delivery. 6 (2015) 1239-1241]. Again, authors must account of the stability pf the prepared liposomal formulation.

Our response: Information regarding liposome stability was added in the section describing liposome characterization and the indicated citation regarding challenges of oral delivery of liposomes has been added.

  1. The conclusion is very poor and must be improved.

Our response: Conclusions have been reformulated to improve their clarity.

Reviewer 3 Report

A well-written paper. Suggest to add more details on the initial PK study (referral to reference 35 could include the name of the scientific team)

Interesting computational approach of the issue.

Author Response

A well-written paper. Suggest to add more details on the initial PK study (referral to reference 35 could include the name of the scientific team)

Our response:

As suggested by the reviewer the description of PK study has been extended.

Round 2

Reviewer 1 Report

Accept in current form.

Author Response

The manuscript has been accepted by the reviewer.

Reviewer 2 Report

Although the authors have made substantial modifications in the revised version of the manuscript, yet the manuscript still suffering from some drawbacks. 

- The abstract needs to be written in the way I mentioned in my previous comments.

- Information mentioned in 92-96 are contradicting. Authors mentioned that liposomes topology remains unchanged sufficiently long to ensure effective absorption. Then mentioned that  liposomes suspension have low stability [33]. This issue was of major concern in the first round of review. 

- Page 5, line 208; authors mentioned an integrated equation to quantify the level of vitamin D in the serum while, the correct is to mention the linear equation.

- The manuscript still suffering from many typo mistakes. For example "ACU1day-30day" in line 288, 289. 

- The manuscript still needs English revision. 

Author Response

The list of reviewer comments together with our corrections

The abstract needs to be written in the way I mentioned in my previous comments.

Our response: The abstract has been changed as indicated by reviewer.

- Information mentioned in 92-96 are contradicting. Authors mentioned that liposomes topology remains unchanged sufficiently long to ensure effective absorption. Then mentioned that  liposomes suspension have low stability [33]. This issue was of major concern in the first round of review. 

Our response: We are sorry for the unfortunate statement. The inconsistency has been removed.

- Page 5, line 208; authors mentioned an integrated equation to quantify the level of vitamin D in the serum while, the correct is to mention the linear equation.

Our response

The error has been corrected.

- The manuscript still suffering from many typo mistakes. For example "ACU1day-30day" in line 288, 289. 

Our response

All abreviations have been checked and if necessary corrected.

- The manuscript still needs English revision. 

Our response: The manuscript has been verified by the qualified person

Round 3

Reviewer 2 Report

No further comments.